# NeMF: Neural Motion Fields for Kinematic Animation

**Chengan He**
Yale University
chengan.he@yale.edu

**Jun Saito**
Adobe Research
jsaito@adobe.com

**James Zachary**
Adobe Research
zachary@adobe.com

**Holly Rushmeier**
Yale University
holly.rushmeier@yale.edu

**Yi Zhou**
Adobe Research
yizho@adobe.com

## Abstract

We present an implicit neural representation to learn the spatio-temporal space of kinematic motions. Unlike previous work that represents motion as discrete sequential samples, we propose to express the vast motion space as a continuous function over time, hence the name *Neural Motion Fields (NeMF)*. Specifically, we use a neural network to learn this function for miscellaneous sets of motions, which is designed to be a generative model conditioned on a temporal coordinate $t$ and a random vector $z$ for controlling the style. The model is then trained as a Variational Autoencoder (VAE) with motion encoders to sample the latent space. We train our model with a diverse human motion dataset and quadruped dataset to prove its versatility, and finally deploy it as a generic motion prior to solve task-agnostic problems and show its superiority in different motion generation and editing applications, such as motion interpolation, in-betweening, and re-navigating. More details can be found on our project page: https://cs.yale.edu/homes/che/projects/nemf/.

## 1  Introduction

Motion synthesis and editing is a core problem in animation and game production, as well as in emerging applications like artificial agents. Traditional algorithms have limited capability of automatically producing convincing motions with high diversity and complexity. With the recent availability of large-scale motion capture data, more interest has shifted to deep learning-based methods. People have adapted various deep learning technologies such as Recurrent Neural Networks [34], Reinforcement Learning [25, 35] and Normalizing Flows [10] for skeletal motion generation. However, those methods are designed as an auto-regressive process that depends on its own past values and some stochastic terms. With this constraint, those methods have to sequentially predict the motion at discrete time steps. Consequently, they cannot just directly infer the motion at certain frames since they must predict the past motion first. Moreover, with this temporally asymmetric design, it is always hard for those methods to incorporate the control or editing of future frames and, thus, it is difficult to apply them to tasks like motion in-betweening.

Inspired by the recent success in neural radiance fields (NeRF) for novel view synthesis [32], we introduce *Neural Motion Fields (NeMF)* to model the spatio-temporal space of kinematic motions. Given that a motion sequence consists of different poses at different time steps, we represent a motion sequence as a continuous function $f : t \mapsto f(t)$ which parameterizes the entire sequence by the temporal coordinate $t$. This function can be approximated by a multilayer perceptron network (MLP) whose parameters are optimized by minimizing the reconstruction loss between the generated and ground truth motion.

36th Conference on Neural Information Processing Systems (NeurIPS 2022).

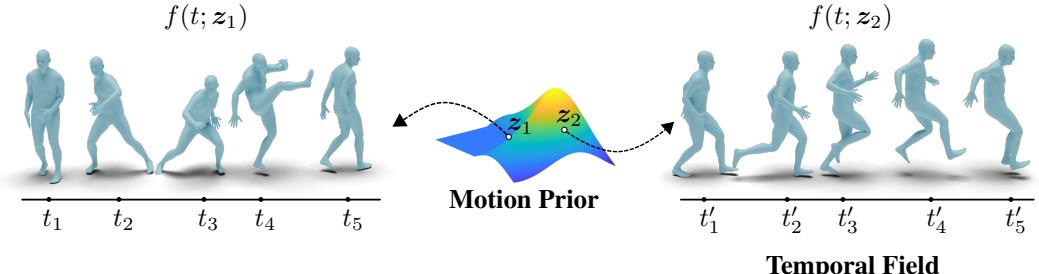

$f(t; \boldsymbol{z}_1)$ $\qquad$ $f(t; \boldsymbol{z}_2)$

$\boldsymbol{z}_1$ $\boldsymbol{z}_2$

**Motion Prior**

$t_1 \quad t_2 \quad t_3 \quad t_4 \quad t_5$ $\qquad$ $t'_1 \quad t'_2 \quad t'_3 \quad t'_4 \quad t'_5$

**Temporal Field**

Figure 1: Overview of generative NeMF, which consists of a motion prior and a continuous temporal field. By sampling the motion prior, NeMF is able to infer a variety of motions with different styles. By sampling the temporal field, NeMF is able to generate motion at arbitrary frames.

However, the NeMF function defined above overfits one motion sequence. To extend it to general motion synthesis, as illustrated in Figure 1, we further introduce a random vector $\boldsymbol{z}$ as the conditioning variable to the function as $f(t; \boldsymbol{z})$, which now encodes the mapping from the normal distribution to the manifold spanned by all plausible motions. By varying $\boldsymbol{z}$, we expect $f$ to output motion with different styles, and by varying $t$, we obtain the pose at arbitrary time. We train this generative NeMF model in the form of a Variational Autoencoder (VAE), where motion sequences are fed to convolutional encoders first to obtain the sampling of $\boldsymbol{z}$, and then $\boldsymbol{z}$ is passed through an MLP decoder for the reconstruction of the whole motion. To the best of our knowledge, our paper presents the first continuous and generative implicit motion representation that can sample different types of motion at an arbitrary time step.

This neural motion representation can serve as a generic motion prior to solve task-agnostic problems. Following Li et al. [21] who trained a hierarchical human motion prior and applied it in different tasks, we demonstrate utilizing a trained NeMF in various applications, including motion interpolation, motion in-betweening and motion re-navigating. We formulate those tasks as optimization problems to find the latent variable $\boldsymbol{z}$ that minimize the target energies and restore the complete plausible motion sequences. Although we did not train this NeMF model for any of these tasks specifically, it achieves performance equivalent or superior to task-specific alternatives in our experiments.

In summary, our contributions are as follows:

- We propose NeMF that represents the continuous motion field as a function over time and design a VAE architecture to train a generative NeMF.
- We validate that our model can reconstruct and synthesize motion with superior quality and diversity than state-of-the-art neural motion priors.
- We demonstrate NeMF in various offline motion editing and creation tasks.

## 2 Related Work

To clarify our design choices, we categorize the kinematic motion modeling into two types: time series models and space-time models. The former approach views motions as the kinematic pose evolving over time, which is often the choice for interactive applications where the future depends on unknown factors such as real-time user inputs. The latter is our approach, which is suitable for offline applications where we know what to expect for the overall motion within some time range.

### 2.1 Time Series Models

Time series models are often formulated as an auto-regressive model that predicts the future based on current and past observations. The predictions are fed into the model again to make further predictions recursively. GPDM [44] modeled such dynamics of human locomotion over time with Gaussian process. Recently people studied various deep learning-based auto-regressive architectures, including Recurrent Neural Networks [5, 29, 49], Reinforcement Learning Networks, Neural ODE [16], Transformers [1, 22, 36] and other attention models [28]. Many works demonstrated success in

real-time interactive kinematic character controls [13, 40–42, 47]. The auto-regressive approach is also suitable for embracing the uncertainty of future predictions with generative models, for example, in the form of VAE as in HuMoR [38] and Motion VAE [25], and in the form of Normalizing Flows as in MoGlow [10].

In applications, Harvey et al. [9] used LSTM [11] with positional encoding to achieve the state-of-the-art result for the motion in-betweening task. But it is limited to a fixed in-betweening setting where some consecutive past frames need to be given as the seed. One of the key applications of our model is also motion in-betweening, but our method has no constraint on the setting and can generate the missing motion both between motion clips and between sparse keyframes.

## 2.2 Space-Time Models

Different from time series models, another approach is to directly model the spatio-temporal kinematic state itself. Such models are often used in combination with a time series model, as in [20] where the dynamic control policy is learned over the GPLVM-based spatial kinematic prior. One of our inspirations, Motion Fields [19], can be seen as a variant of this approach where the time-series dynamics are learned over the spatial kinematic model represented directly by the data and a hand-crafted distance function. More recently, space-time neural network models overcame the scalability issue in previous methods due to the availability of large-scale datasets [12, 14, 15, 45, 48]. Zhou et al. [51] demonstrated long-term motion in-betweening by learning the space-time kinematic prior without supervision using Generative Adversarial Networks (GAN) [7]. Kaufmann et al. [18] also proposed a convolutional network for motion in-betweening. Our other inspiration, HM-VAE [21], learned a generic motion prior for multiple downstream applications. However, their reliance on a temporally convolutional decoder fixes the output frame rate, thus limiting its application to temporal sub-sampling. As a space-time model, our work NeMF has the advantage of being able to edit any frames in the motion sequence while maintaining fidelity and the source style compared to the aforementioned works.

## 2.3 Implicit Neural Representations

Implicit neural representations explosively gained popularity with the success of SIREN [39] and NeRF [32], which achieved state-of-the-art results in many tasks including novel view synthesis, geometry reconstruction, and solving differential equations. Among different tasks, the core idea of neural implicit representations is to build a *continuous function* that maps the spatial or temporal coordinates to any signal at that location while maintaining high-frequency details. Inspired by their success, our key insight is to interpret time as the parameter of a *motion function* and learn the landscape of the spatio-temporal kinematics manifold as an implicit motion representation parameterized by temporal coordinates. Similar ideas have been proposed in modeling time-varying 3D geometries [33] and dynamic scenes [24, 37], and we extend it to the animation domain.

# 3 Formulation

In this section, we will first give the definition and notation of our motion representation, then present the formulation of NeMF for a single motion sequence, followed by an extended version of generative NeMF for the entire motion space.

## 3.1 Motion Representation

Similar to Zhou et al. [51] and Li et al. [21], we divide the motion into two parts: *local motion*, which contains the pose of the skeleton relative to the root at time $t$, and *global motion*, which is the global translation of the root joint. Following Fussell et al. [6], we represent the local motion at time $t$ as a matrix $\mathbf{X}_t$ composed of joint positions $\mathbf{x}_t^p \in \mathbb{R}^{J \times 3}$, velocities $\dot{\mathbf{x}}_t^p \in \mathbb{R}^{J \times 3}$, rotations $\mathbf{x}_t^r \in \mathbb{R}^{J \times 6}$ in 6D rotation form [50], and angular velocities $\dot{\mathbf{x}}_t^r \in \mathbb{R}^{J \times 3}$:

$$\mathbf{X}_t = (\mathbf{x}_t^p \quad \dot{\mathbf{x}}_t^p \quad \mathbf{x}_t^r \quad \dot{\mathbf{x}}_t^r) \in \mathbb{R}^{J \times 15}, \tag{1}$$

where $J$ is the number of joints. Since all other quantities can be computed from joint rotations, we focus on predicting joint rotations $\mathbf{x}_t^r$ in the formulation. We then factor out the root orientation $\mathbf{r}_t^o \in \mathbb{R}^6$ from local motion by multiplying the inverse of the root transform to each quantity in $\mathbf{X}_t$.

In this way, all poses will be transformed to a local space with the same facing direction, thus making them easier to learn for our network. For the global motion, based on previous works [21, 51], we use a neural network to predict the velocity of the root joint $\dot{\mathbf{r}}_t \in \mathbb{R}^3$ as well as its height $\mathbf{r}_t^h \in \mathbb{R}$ from $\mathbf{X}_t$. We provide details of our global motion predictor in the supplemental.

## 3.2 Neural Motion Fields

Unlike most other works that represent motion as a discrete sequential process, we represent motion as a continuous vector field of kinematic poses in the *temporal* domain. Hence we define this motion field as a function that maps temporal coordinates $t$ to joint rotations and root orientations:

$$f : t \mapsto (\mathbf{x}_t^r, \mathbf{r}_t^o).\tag{2}$$

The function $f$ can be approximated by a neural network with the parameters $\theta$ that can be optimized by minimizing the reconstruction loss between the generated and ground truth motion, where the ground truth poses can be considered as discrete samples of $f$ at integer time steps.

Similar to NeRF [32], we train an MLP with positional encoding of $t$ to fit a given motion sequence. Considering a sequence with $T$ frames, we first convert the 6D rotations $\mathbf{x}_t^r$ and $\mathbf{r}_t^o$ to rotation matrices $\mathbf{R}_t$ and $\mathbf{R}_t^o$ with the Gram-Schmidt-like process described in [50], and then compute the geodesic distance to measure the rotational difference:

$$\mathcal{L}_{\text{rot}} = \sum_{t=1}^{T} \arccos \frac{\text{Tr}\left(\mathbf{R}_t(\hat{\mathbf{R}}_t)^{-1}\right) - 1}{2}, \quad \mathcal{L}_{\text{ori}} = \sum_{t=1}^{T} \arccos \frac{\text{Tr}\left(\mathbf{R}_t^o(\hat{\mathbf{R}}_t^o)^{-1}\right) - 1}{2}.\tag{3}$$

We then evaluate the $L_1$ loss on local joint positions obtained from forward kinematics (FK) [43], which regularizes the skeletal topology to obtain better results [34]:

$$\mathcal{L}_{\text{pos}} = \sum_{t=1}^{T} \|\mathbf{x}_t^p - \hat{\mathbf{x}}_t^p\|_1.\tag{4}$$

The reconstruction loss is finally expressed as a weighted sum of the above terms with weighting factors $\lambda_{\text{rot}}$, $\lambda_{\text{ori}}$, and $\lambda_{\text{pos}}$:

$$\mathcal{L}_{\text{rec}} = \lambda_{\text{rot}}\mathcal{L}_{\text{rot}} + \lambda_{\text{ori}}\mathcal{L}_{\text{ori}} + \lambda_{\text{pos}}\mathcal{L}_{\text{pos}},\tag{5}$$

and we optimize the network parameters $\theta$ to minimize this loss function.

## 3.3 Generative Neural Motion Fields

Although the neural motion field defined above can represent a motion sequence in a compact way, it cannot generate a variety of motions. To change to a different motion sequence, the entire network needs to be re-trained from scratch, which severely limits its application. Therefore, in this section, we extend NeMF to a generative model which can represent the entire motion space instead of a specific motion sequence.

First of all, we introduce a conditioning variable $\mathbf{z}$ to the input of $f$, thus parameterizing the entire *spatio-temporal* kinematics space as a function:

$$f : (t, \mathbf{z}) \mapsto (\mathbf{x}_t^r, \mathbf{r}_t^o),\tag{6}$$

where $\mathbf{z}$ defines the spatial location on the manifold and $t$ controls the temporal evolution of the sequence.

We then propose a VAE to formulate Equation 6 as a latent variable model with Gaussian distribution. Based on our motion representation, the VAE contains two separate convolutional motion encoders to learn and parameterize the posterior distribution of latent variables $\mathbf{z}_l$ and $\mathbf{z}_g$, which control the local motion and root orientation respectively:

$$q_{\phi_1}(\mathbf{z}_l \mid \mathbf{X}) = \mathcal{N}(\mathbf{z}_l; \mu_{\phi_1}(\mathbf{X}), \sigma_{\phi_1}(\mathbf{X})), \quad q_{\phi_2}(\mathbf{z}_g \mid \mathbf{r}^o) = \mathcal{N}(\mathbf{z}_g; \mu_{\phi_2}(\mathbf{r}^o), \sigma_{\phi_2}(\mathbf{r}^o)),\tag{7}$$

where $\mathbf{X}$ and $\mathbf{r}^o$ are the concatenation of all $\mathbf{X}_t$ and $\mathbf{r}_t^o$ within the same sequence.

The combination of $\mathbf{z}_l$ and $\mathbf{z}_g$ forms the final representation of the latent variable $\mathbf{z}$, which are then passed through the MLP decoder to produce joint rotations $\mathbf{x}_t^r$ and root orientations $\mathbf{r}_t^o$ for each time step $t$, thus, defining the output probability distribution $p_\theta(\mathbf{x}^r, \mathbf{r}^o \mid \mathbf{z}_l, \mathbf{z}_g)$.

In addition to the probabilistic perspective, our model can also be interpreted as learning a *motion prior* from input sequences, where each latent variable on the prior corresponds to a natural motion sequence. The MLP decoder then approximates the mapping function between the motion prior and pose space, thus allowing the navigation in the pose space to have a consistent motion style with a fixed $\boldsymbol{z}$.

During training, we consider the modified variational lower bound:

$$\log p_\theta(\mathbf{x}^r, \mathbf{r}^o) \geq \mathbb{E}_{q_{\phi_1}, q_{\phi_2}}[\log p_\theta(\mathbf{x}^r, \mathbf{r}^o \mid \boldsymbol{z}_l, \boldsymbol{z}_g)] - \mathcal{D}_{\mathrm{KL}}(q_{\phi_1}(\boldsymbol{z}_l \mid \mathbf{X}) \parallel p(\boldsymbol{z}_l)) \\ - \mathcal{D}_{\mathrm{KL}}(q_{\phi_2}(\boldsymbol{z}_g \mid \mathbf{r}^o) \parallel p(\boldsymbol{z}_g)), \tag{8}$$

where the expectation term measures the reconstruction error of the decoder, and the KL divergence $\mathcal{D}_{\mathrm{KL}}$ regularizes the encoders' outputs to be near $p(\boldsymbol{z}_l)$ and $p(\boldsymbol{z}_g)$, which are $\mathcal{N}(\mathbf{0}, \mathbf{I})$ in our scenario. Thus, we formulate the loss function $\mathcal{L} = \mathcal{L}_{\mathrm{rec}} + \lambda_{\mathrm{KL}}\mathcal{L}_{\mathrm{KL}}$ with the weight $\lambda_{\mathrm{KL}}$ to approximate Equation 8, and the network parameters $(\phi_1, \phi_2, \theta)$ are optimized during training to minimize the loss.

### 3.4 Applications

After training the generative NeMF, we can deploy it as a generic motion prior to solve different tasks via latent space optimization.

**Motion In-betweening** is a long-standing animation creation problem of generating motion in the interval between two clips or sets of keyframes. Our insight is, given a sparse set of observations, we can search in the latent space of NeMF to approximate the entire underlying motion thanks to our continuous representation. Thus, we define the energy function as the reconstruction loss on given frames $\mathcal{T}$, which measures the differences on joint rotations, root orientations, joint positions and root translations:

$$\boldsymbol{z}_l^*, \boldsymbol{z}_g^* \coloneqq \underset{\boldsymbol{z}_l, \boldsymbol{z}_g}{\arg\min} \sum_{\mathcal{T}} \lambda_{\mathrm{rot}}\mathcal{L}_{\mathrm{rot}} + \lambda_{\mathrm{ori}}\mathcal{L}_{\mathrm{ori}} + \lambda_{\mathrm{pos}}\mathcal{L}_{\mathrm{pos}} + \lambda_{\mathrm{trans}}\mathcal{L}_{\mathrm{trans}}, \tag{9}$$

where $\mathcal{L}_{\mathrm{trans}}$ weighted by $\lambda_{\mathrm{trans}}$ evaluates the $L_1$ loss on root joint positions $\mathbf{r}_t$ predicted from our standalone global motion predictor. To facilitate convergence, SLERP is first used to perform interpolation in the joint angle space, and then the interpolated inputs are passed through encoders to obtain the initialization of $\boldsymbol{z}_l$ and $\boldsymbol{z}_g$.

**Motion Re-navigating** is a task where we redirect a reference motion with a new trajectory while preserving its style. More formally, the goal is to generate a motion that 1) looks similar to the given exemplar; and 2) follows the given trajectory as close as possible. To formulate the energy function, we first borrow ideas from time-series analysis by introducing the soft dynamic time warping metric ($\mathcal{L}_{\mathrm{sDTW}}$) [4] to measure the similarity between joint positions in the canonical frame [38]. Then, we project the predicted root joints onto the ground plane and compute the $L_1$ loss between them and the given 2D trajectory. Last but not least, an additional regularization term is introduced to ensure that the generated and reference motion have a similar angle between their forward direction $\mathbf{f}_t$ and the tangent direction $\mathbf{t}_t$ on the trajectory, where the similarity is determined by $\mathcal{L}_{\mathrm{sDTW}}$ as well. In total, the energy function we try to minimize can be expressed as:

$$\boldsymbol{z}_l^*, \boldsymbol{z}_g^* \coloneqq \underset{\boldsymbol{z}_l, \boldsymbol{z}_g}{\arg\min} \sum_{t=1}^T \lambda_{\mathrm{sim}}\mathcal{L}_{\mathrm{sDTW}}(\mathbf{x}_t^p, \hat{\mathbf{x}}_t^p) + \lambda_{\mathrm{traj}}\|\mathbf{r}_t^{\mathrm{proj}} - \hat{\mathbf{r}}_t^{\mathrm{proj}}\|_1 + \lambda_{\mathrm{angle}}\mathcal{L}_{\mathrm{sDTW}}(\mathbf{f}_t \cdot \mathbf{t}_t, \hat{\mathbf{f}}_t \cdot \hat{\mathbf{t}}_t), \tag{10}$$

where $\lambda_{\mathrm{sim}}$, $\lambda_{\mathrm{traj}}$ and $\lambda_{\mathrm{angle}}$ are three weighting factors introduced to balance different terms. In our experiments, $\boldsymbol{z}_l$ are initialized from the encoder's output, while $\boldsymbol{z}_g$ are optimized from scratch.

## 4 Experiments

We train our model on the AMASS dataset [27] for most of the experiments. After processing, we have roughly 20 hours of human motion sequences at 30 fps for training and testing. We additionally train our model for the reconstruction experiments on a quadruped motion dataset [47], which contains 30 minutes of dog motion capture at 60 fps. Details are provided in the supplemental.

### 4.1 Sanity Test

#### 4.1.1 Motion Reconstruction

We first perform a sanity test to validate our single-motion NeMF model described in Section 3.2, where we test the reconstruction capability of NeMF with different lengths of motions.

For AMASS data, we pick 16 different motion sequences for each length and report the reconstruction errors in Table 1. For evaluation metrics, we report mean rotation error (**MRE**, °) and mean position error (**MPE**, $cm$) for each joint, which measure the geodesic distance between joint rotations and Euclidean distance between root-aligned joint positions. For the root joint, we further report its orientation error (**MOE**, °), which measures the geodesic distance on the root orientation. From the results shown in Table 1, we can observe that a simple MLP with positional encoding is able to to achieve very low reconstruction error (lower than $4mm$ positional error and $0.7°$ rotational error) for sequence lengths varying from 32 to 512 frames.

We further train our NeMF model on a very long ($4,336$ frames and $73s$) quadruped motion sequence and visualize the result in the supplemental. Our predicted motion is visually almost identical to the ground truth.

#### 4.1.2 Temporal Sampling

Unlike other motion models, NeMF is theoretically guaranteed to generate smooth motion in arbitrary frame rates. We find that the dimension of the Fourier features generated by positional encoding plays a critical role in the generalization here.

Since the maximal frequency of the Fourier features is determined by a hyperparameter $L$ as illustrated in [32], we uniformly sample $L$ from 1 to 21 and train NeMF with these different setups. For each trained model, we respectively generate 30 and 60 fps motions and report their mean per-joint velocity as the smoothness metric in Figure 2. Although a large value of $L$ does not harm the results with the training frame rate, such a model struggles to generalize to different frame rates. However, since positional encoding is crucial to achieve satisfactory reconstruction results, we set $L = 7$ throughout our experiments to balance the trade-off. In the supplemental video, we show that the generated motion remains smooth even sampled at 240 fps.

Table 1: Mean reconstruction errors of single-motion NeMF for motion of different lengths. Mean rotation error (°), mean position error ($cm$), and mean orientation error (°) are reported.

| | Sequence Lengths | | | | |
|---|---|---|---|---|---|
| Metrics | 32 | 64 | 128 | 256 | 512 |
| MRE | 0.610 | 0.482 | 0.381 | 0.369 | 0.379 |
| MPE | 0.314 | 0.249 | 0.213 | 0.192 | 0.170 |
| MOE | 0.465 | 0.340 | 0.344 | 0.321 | 0.300 |

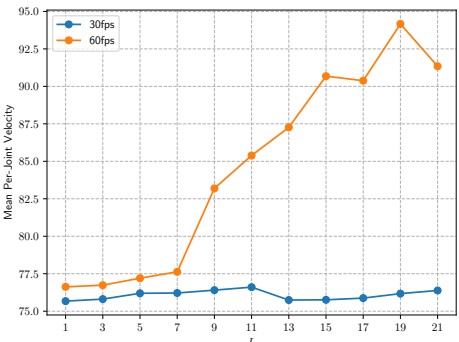

Figure 2: Mean per-joint velocity ($cm/s$) evaluated on the same motion with different $L$.

### 4.2 Generative NeMF

#### 4.2.1 Evaluation Metrics

We evaluate the reconstruction capability of our generative NeMF model through direct network inference and motion synthesis capability through latent space sampling. To better describe motion reconstruction, we further measure the Euclidean distance on the root joint translation (**MTE**, $cm$) and global joint displacement on the final step (**FDE**, $cm$) [45]. For estimating the versatile motion quality, we introduce three metrics, namely Fréchet Inception Distance (**FID**), diversity (**Diversity**), and foot skating (**FS**), following related papers [8, 25, 36, 47]. Generally speaking, a lower FID suggests a more natural result, a higher diversity indicates a more various result, and foot skating shows the accumulated drift of foot joints during contact. Please see our supplemental for details.

### 4.2.2 Ablation Study

We first take a look at the effect of the positional encoding function. By removing positional encoding, the input temporal coordinate $t$ will be passed directly to the MLP to infer pose parameters. Results in the first row of Table 2 show that the model without positional encoding produces higher errors in most metrics we evaluate, thus demonstrating the necessity of positional encoding in terms of the fidelity of the results.

We further experiment on using single motion encoder instead of two separated ones to process both local motion and root orientation. In this simpler design, the root orientation and local motion are entangled in the same latent space. From the results we reported in the second row of Table 2, though the simpler version produces a lower reconstruction error on the root orientation, it has worse performance on joint rotations, joint positions and root translations.

Table 2: Ablation study.

| | Motion Reconstruction | | | | Motion Synthesis | | |
|---|---|---|---|---|---|---|---|
| | **MRE** ↓ | **MPE** ↓ | **MOE** ↓ | **MTE** ↓ | **FID** ↓ | **Diversity** ↑ | **FS** ↓ |
| No Positional Encoding | 6.413 | 2.952 | 6.027 | 33.524 | 2.146 | 2.783 | 0.612 |
| Single Motion Encoder | 8.661 | 4.503 | **5.982** | 35.391 | 2.185 | **2.806** | 0.818 |
| Full Model | **5.988** | **2.870** | 6.157 | **29.692** | **2.073** | 2.774 | **0.573** |

### 4.2.3 Latent Space Sampling

In our model, we can navigate in the latent space to control the motion styles and synthesize novel motion. To examine the smoothness of the latent space and see whether our model can blend different styles of motion at the sequence level, we linearly interpolate $z$ from two existing motion sequences and infer the novel ones. Previous works like [46] demonstrate motion interpolation between similar motion patterns, such as from walking to zombie-style walking. However, we show in the supplemental video that our model can blend the high-level perceptual style between much harder cases like jumping and punching.

Since we disentangle the latent space for local motion and root orientation, we also experiment to combine $z_l$ and $z_g$ from different motions. Through this, we can create interesting editing results like canceling the spinning motion of a pirouette jump (see supplemental).

### 4.2.4 Comparison with Other Generative Models

We compare our method with other deep learning-based motion priors on motion reconstruction and synthesis in Table 3. We select HuMoR [38] and HM-VAE [21] to represent the "time series model" and "space-time model" respectively.

Table 3: Comparison of NeMF with other generative motion models.

| | Motion Reconstruction | | | | | Motion Synthesis | | |
|---|---|---|---|---|---|---|---|---|
| | **MRE** ↓ | **MPE** ↓ | **MOE** ↓ | **MTE** ↓ | **FDE** ↓ | **FID** ↓ | **Diversity** ↑ | **FS** ↓ |
| HuMoR [38] | 13.008 | 9.071 | 17.097 | **23.882** | 62.756 | 8.687 | 1.741 | 0.904 |
| HM-VAE [21] | 10.258 | 7.686 | 13.054 | 90.924 | 137.218 | 7.998 | 2.002 | 0.690 |
| Ours | **5.988** | **2.870** | **6.157** | 29.692 | **42.985** | **6.508** | **2.118** | **0.566** |

Although HuMoR can generate convincing motions, we observe two shortcomings of HuMoR: 1) in motion reconstruction, HuMoR's results will gradually diverge due to its auto-regressive prediction. From the quantitative results in Table 3 and qualitative results in Figure 3, divergence can be observed in both the large FDE and global translation difference respectively. 2) In motion synthesis, HuMoR tends to favor common motions like walking when inferring long sequences, thus yielding a relatively high FID and low diversity. As for our method, we can get rid of these artifacts since we handle the entire sequence at once instead of predicting the motion frame by frame.

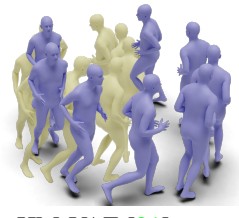 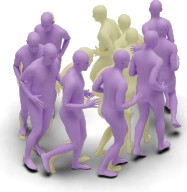 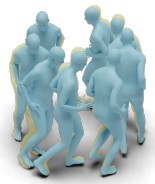

|        HM-VAE [21]        |        HuMoR [38]        |        Ours        |

Figure 3: Comparison of motion reconstruction with other generative motion models. Predicted motions are overlapped on top of the ground truth motion (yellow).

Compared to HuMoR and our model, HM-VAE models an over-smoothed latent space, thus filtering out high-frequency details in both the reconstructed and synthesized motions. In short, our model outperforms these state-of-the-art methods in most of the metrics reported in Table 3, as well as achieving the closest match of the ground truth motion visualized in Figure 3.

## 4.3 In-betweening Tasks

In this task, we compare our method with traditional and deep learning-based motion in-betweening methods. For quantitative evaluation, we use the FID and foot skating metrics only, since reconstruction errors cannot fully reflect the quality of motion for highly-varied plausible solutions.

Table 4: Motion clips in-betweening.

|  | **FID** | | |
| --- | --- | --- | --- |
| Length (frames) | 10 | 20 | 30 |
| SLERP | 0.027 | 0.170 | 0.455 |
| Inertialization [2] | 0.025 | 0.184 | 0.496 |
| RMI [9] | 0.264 | 0.362 | 0.609 |
| HM-VAE [21] | 0.137 | 0.387 | 0.675 |
| Ours | **0.024** | **0.141** | **0.365** |
| | **Foot Skating** | | |
| SLERP | 1.193 | 1.200 | 1.023 |
| Inertialization [2] | 1.237 | 1.234 | 1.151 |
| RMI [9] | 1.629 | 1.833 | 1.643 |
| HM-VAE [21] | 0.862 | 0.835 | 0.832 |
| Ours | **0.646** | **0.697** | **0.660** |

Table 5: Sparse keyframe in-betweening.

|  | **FID** | | | |
| --- | --- | --- | --- | --- |
| Length (frames) | 5 | 10 | 15 | 20 |
| SLERP | **0.032** | 0.305 | 0.713 | 1.191 |
| HM-VAE [21] | 2.300 | 2.370 | 2.434 | 2.423 |
| Ours | 0.085 | **0.302** | **0.612** | **0.879** |
| | **Foot Skating** | | | |
| SLERP | 0.868 | 1.224 | 1.278 | 1.193 |
| HM-VAE [21] | 0.892 | 0.837 | 0.841 | **0.751** |
| Ours | **0.719** | **0.826** | **0.837** | 0.804 |

**Motion Clips In-betweening.** In this task, we aim at generating motion in the interval from 10 frames ($0.33s$) to 30 frames ($1s$) between two clips. We compare with traditional local interpolation methods including SLERP and Inertialization [2], and deep learning-based methods including Robust Motion In-betweening (RMI) [9] and HM-VAE [21]. In the results reported in Table 4, our method outperforms all other alternatives quantitatively. We further experiment on real dancing footage by randomly picking several pairs of videos from AIST++ [23] with their corresponding SMPL [26] parameters. We set the transition length to 30 frames and optimize for a latent variable to produce the gap-filling motion. As demonstrated in Figure 4, our model is capable of producing a natural one-second transition between these real dancing footage.

**Sparse Keyframe In-betweening.** In this task, a set of sparse keyframe skeleton poses are given every 5, 10, 15 or 20 frames. We compare our method with SLERP and HM-VAE, and report the quantitative results in Table 5. Neither RMI nor Inertialization is applicable here since they both require multiple frames at the beginning.

From Table 5, SLERP has the best FID score for interval of 5 frames, but becomes worse with increased interval length. Compared to SLERP and HM-VAE, ours has much better quantitative and qualitative results, especially for long intervals (Figure 5). In the onionskin images, though SLERP's

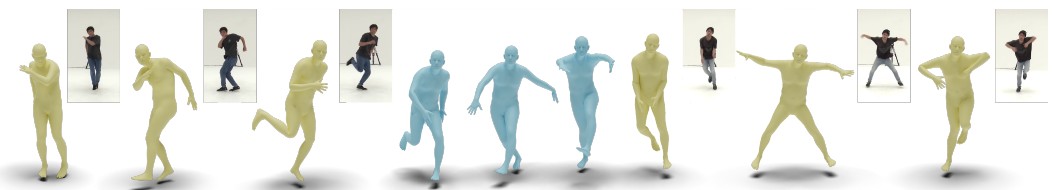

Figure 4: Generating the transition (cyan) between AIST++ motion clips (yellow).

result looks similar to ours, our result has more natural high-frequency details. HM-VAE fails to reconstruct the poses at keyframes and generates over-smoothed results. Our result, although different from the ground truth, is visually plausible as in the supplemental video.

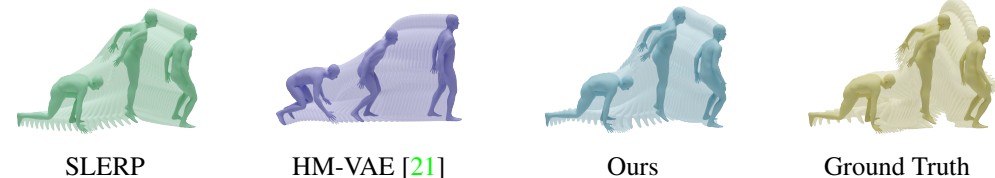

| SLERP | HM-VAE [21] | Ours | Ground Truth |

Figure 5: Comparison of sparse keyframe in-betweening. The translucent in-between poses are generated from the opaque reconstructed keyframes given every 20 frames.

## 4.4 Re-navigating Tasks

In this task, we experiment on redirecting the reference motion to different synthetic trajectories. Qualitative results are reported in Figure 6. The original motion style is well preserved in the results, and the character has a natural orientation while walking along the trajectory.

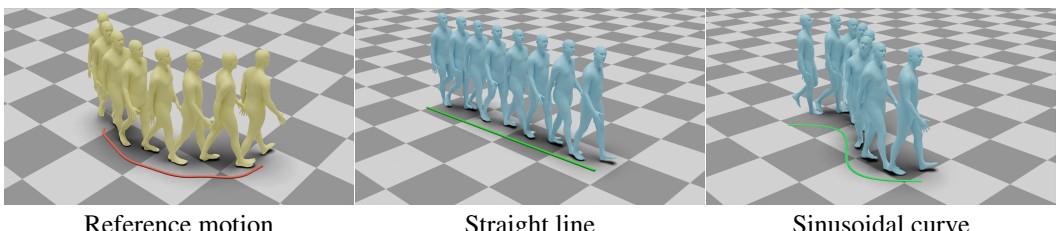

| Reference motion | Straight line | Sinusoidal curve |

Figure 6: Motion re-navigating from a reference walking motion (left) to a straight line (middle) and sinusoidal curve (right).

## 5 Conclusion

We propose an implicit neural motion representation that defines a continuous motion field over style and time. We design it to be a generative model for the whole motion space with a learned prior. We design and test different network architectures and use the trained generative model as a motion prior for solving different tasks like motion interpolation, motion in-betweening and motion re-navigating.

## 5.1 Limitation and Future Work

In this paper we train the generative NeMF model in the form of a VAE, so it has the limitation of not always giving satisfactory results when sampling the latent space. A promising direction is to design other generative architectures such as GAN, Normalizing Flows, and Diffusion Models for NeMF. Besides, though several latent variables may exist that satisfy the optimization constraints, our deterministic optimization process can only find one possible $z$ as the final result, thus limiting probabilistic motion synthesis like MoGlow [10].

In the future, since the optimization problems we formulated are analogous to the latent space optimization problems of StyleGAN [17], how to transfer the algorithm of those successful applications designed for StyleGAN to our NeMF setup is an interesting direction to explore.

So far our method focuses on motion modeling and cannot adapt to outputs with different body shapes. On the other hand, some works [3, 30, 31] leverage implicit representations to model skinned articulated objects with given poses. Therefore, it would be interesting to combine our work with them to enable the animation and rendering of a specific character with novel motions.

To keep our model task-agnostic, we choose to solve motion tasks by optimizing $z$. But for practical applications, task-specific inference models may be preferred for performance consideration. To have an end-to-end inference model for real-time applications, a possible solution is to design and train the encoder to directly predict $z$ with different input settings.

### 5.2 Broader Impact

Our model can be applied for kinematic motion creation in animation and game production. Due to the limitation of diversity in the training data, our model is not guaranteed to generate feasible but uncommon motions. We cannot guarantee that our model will avoid generating motions that could be perceived as offensive to some viewers.

## Acknowledgments and Disclosure of Funding

This work was supported in part by NSF Grant No. IIS-2007283. Many thanks to Ruben Villegas for helpful discussions. Special thanks to Jiaman Li for sharing with us the pre-trained HM-VAE model.

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
