# OpenReview forum: "NeMF: Neural Motion Fields for Kinematic Animation"
_NeurIPS.cc/2022/Conference — NeurIPS 2022 Accept_

### Official Review · Reviewer_Qiic · 2022-07-09

**Rating:** 8
**Confidence:** 3
**Soundness:** 4 excellent
**Presentation:** 4 excellent
**Contribution:** 3 good

**Summary:**

The authors propose to formulate human motion as a generative implicit representation. The authors demonstrate the effectiveness of such approach. The authors showcase the model in various tasks.

**Questions:**

I don't have any specific technical question.

**Limitations:**

The authors do a great job in this aspect.

**Strengths And Weaknesses:**

Strength:
* Overall, I love everything in the paper. Things are presented in a clear and coherent way, making things easy to follow.
* The baselines and experiment comparisons are all very solid. The authors did a great job explaining the baselines and comparing with them.
* The authors did an excellent job explore using the model in various scenarios. This is very informative and showcases the robustness of the model.

Weakness:
* Important related work missing: Occupancy Flow: 4D Reconstruction by Learning Particle Dynamics, ICCV 2019. The part where the authors talk about " Inspired by their success, our key insight is to interpret time as the parameter of a motion function and learn the landscape of the spatio-temporal kinematic manifold as an implicit motion field parameterized by temporal coordinates." This is exactly what the highly-cited prior work has done. Though the prior work differs in motion representation, it should be discussed.
* And a nitpick for the previous point's quote is "implicit motion field" should be "implicit motion representation".

---

> ### Author Response · Authors · 2022-08-01
> **Respone to Reviewer Qiic**
>
> Dear Reviewer Qiic:
>
> Thanks for your review and positive feedback! We will add this important related work and go through our paper carefully to fix some typos and make some descriptions clearer!

---

### Official Review · Reviewer_1jgD · 2022-07-10

**Rating:** 8
**Confidence:** 5
**Soundness:** 4 excellent
**Presentation:** 4 excellent
**Contribution:** 4 excellent

**Summary:**

In this paper, the authors propose a novel representation of human motion as neural motion fields. Inspired by the success of neural fields in other areas, this work learns an implicit function to represent motions. This representation can help to sample diverse and natural motions and facilitate various downstream applications. The experiments demonstrates the ability of their method.

**Questions:**

Please refer to weakness.

**Ethics Review Area:**

["I don’t know"]

**Limitations:**

Please refer to weakness.

**Strengths And Weaknesses:**

Strengths:
+The authors proposed an INR-based model to learn motions. The model is continuous and can generate sequences of arbitrary frame rates in principal.
+Non-autoregressive generation. The function formulation allows the generation of poses at specific time steps directly without first generating frames before them while preserving the smoothness of the full sequence.
+The paper gives extensive experiments on the efficacy and applications of its proposed model.

Weaknesses:
-The formulation of the VAE is very similar to HM-VAE. The introduction of INR is novel and very interesting, but it seems the combination of the two is simple.
-The formulation only uses a latent z to decide the motion type, so for each task the optimization process only finds a deterministic z, which gives a single generated sequence.
-In Figure. 5 the keyframes given for the HM-VAE example are different from the others. Some typos remain in the paper (Section 4.1.1 for example).

---

> ### Author Response · Authors · 2022-08-01
> **Response to Reviewer 1jgD**
>
> Dear Reviewer 1jgD:
>
> Thanks for your review and positive feedback! We will take a closer look at our paper and fix the remaining typos. In the following, we seek to address your concerns one by one:
>
> **Q:** *“The formulation of the VAE is very similar to HM-VAE. The introduction of INR is novel and very interesting, but it seems the combination of the two is simple.”*
>
> **A:** The main purpose of our paper is to introduce the idea of neural motion fields and the potential applications as a motion prior. The VAE is a simple formulation we adopted to generalize the neural motion field to the whole motion space. As we wrote on [L311], more advanced designs such as GAN or Diffusion Models for training the latent space can be left for future work.
>
> ---
>
> **Q:** *“The formulation only uses a latent $z$ to decide the motion type, so for each task the optimization process only finds a deterministic $z$, which gives a single generated sequence.”*
>
> **A:** This is indeed a limitation of our formulation. In our motion prior, there definitely exist multiple latent codes that satisfy the optimization constraints such as motion in-betweening. However, since the optimization process is deterministic, we can only find one possible $z$ as the result. We will mention probabilistic motion synthesis as our limitation in the revision.
>
> ---
>
> **Q:** *“In Figure. 5 the keyframes given for the HM-VAE example are different from the others.”*
>
> **A:** Sorry for the misleading. The opaque frames we visualized are not the given keyframes, but the poses generated on those keyframes. HM-VAE failed to faithfully recover those keyframes, that’s why their visualization looks different from others. It also supports our conclusion on [L273] that “HM-VAE models an over-smoothed latent space”. We will change our wording to demonstrate it more clearly.

---

> > ### Comment · Reviewer_1jgD · 2022-08-08
> > **Comments**
> >
> > Very thanks for the repose. I think all the things in this paper are great. I love the idea of this paper and I believe it is an important progress and baseline for human motion synthesis.

---

### Official Review · Reviewer_wNce · 2022-07-12

**Rating:** 7
**Confidence:** 4
**Soundness:** 4 excellent
**Presentation:** 3 good
**Contribution:** 3 good

**Summary:**

This paper proposes an implicit neural model for kinematic motions. Compared to the seminal NeRF model, which learns a continuous function over spatial coordinates, the proposed NeMF framework learns a continuous function over temporal coordinates. Meanwhile, NeMF outputs a vector of the kinematic pose compared to the color and occupancy information in NeRF. The paper also proposes to extend NeMF beyond only encoding a single motion sequence - following the recent trend on the generalization of implicit neural models. This is achieved by adopting a VAE framework to make NeMF a generative model controlled by a latent code input. The model is trained following the standard VAE training objective including the reconstruction loss and KL loss. The proposed approach is evaluated on several motion generation task (i.e. motion in-betweening and re-navigation) using the AMASS dataset, and compared with several prior methods including HuMoR and HM-VAE. Results have shown that NeMF have achieved better performance on motion reconstruction and motion synthesis (Tab. 3, 4, and 5).

**Questions:**

### Questions
- In general I'm wondering how the latent code handles translation in time. For example, in the current framework, the exact motion clip (e.g. with a time interval [0, T]) after translation (e.g. to interval [-T/2, +T/2]) will have a completely different latent code. This seems a bit awkward and somewhat redundant. I wonder whether there has been any observed pattern on the latent code for translation in time, also, whether one should eventually decouple the translation dimension from the latent code.
- [Eq. 5] Should there also be a loss term for global root translation? The supplementary Sec. 1.1 mentions a loss function for the global motion predictor (i.e. $\mathcal{L}$ in Eq. 2). Is that also part of $\mathcal{L}_{rec}$ here? If not, how is the global motion predictor loss applied in the framework?
- [Tab. 3] How come some of the numbers in "Ours" is exact to those in Tab. 2 (e.g. 5.988 MRE, 2.870 MPE, 6.157 MOE, and 29.692 MTE) while some are different (e.g. 6.508 versus 2.073 FID, 2.118 versus 2.774 Diversity, and 0.566 versus 0.573 FS)?

### Suggestions
- [L171] "In our experiments, $z_l$ and $z_g$ are initialized using the SLERP data to facilitate convergence." -> Detail is unclear. Is SLERP first used to perform interpolation in the joint angle space, and then the joint angles are passed through the encoder to obtain the latent code?
- [Fig. 2] I wonder whether the absolute velocity values are the best metric to look at for this motion reconstruction evaluation. Would it be better to evaluate on the error (of both position and velocity) rather than the absolute value? The issue may be that it is not possible to get the ground-truth position and velocity at frame rates higher than 30 FPS. Then would it be reasonable to train on 15 FPS and evaluate on 30 FPS then just for this experiment?
- [L273] "Compared to HuMoR and our model, it is **obvious** that HM-VAE" models an over-smoothed latent space, thus filtering high-frequency details in both the reconstructed and synthesized motions, ..." -> Not so obvious. Following the weakness point for [20] above, it is unclear how [20] works and why it "models an over-smoothed latent space" while HuMoR and "Ours" don't.
- [L274] "..., making their results worse in both the reconstruction and synthesis task." -> Not precisely. HM-VAE seems to outperform HuMoR sometimes, e.g. on MRE, MPE, MOE?
- [High level] So far the paper still utilizes a low dimensional vector at the output for obtaining human pose and shape. How does it compare to making the model "fully implicit", i.e. on not just the temporal but also the spatial domain (x, y, z) as well, like in the original NeRF and other implicit shape models? I guess as the first step, it may just be easier to rely on a forward function to generate the mesh from the pose vector, which is why the authors go in this route first. Yet a "full implicit" model will make the framework looks more elegant and potentially better integrated with the efforts on the NeRF side.
- [High level] The paper has shown results on an array of "offline" tasks, where the motion of the full sequence can be pre-solved through a latent code and fixed thereafter. Meanwhile, I would imagine an "online/interactive" variant of the NeMF framework would be useful, especially in many gaming applications (e.g. real-time character control, etc.). Are there ways to adapt the current framework to an interactive setup? Perhaps one can run the latent space optimization and motion generation in a continuous loop by streaming motion generation in short time segments (e.g. 1 sec)? If so, it would be nice to add some mention to such extensions.


**Limitations:**

Limitations and potential negative societal impact are addressed in Sec 5.1 and 5.2 respectively.

**Strengths And Weaknesses:**

### Strengths
- [Original] The proposed framework does look novel - by adapting the impactful NeRF model to the domain of kinematic motions. It is certainly interesting to see how these models work not just on static scene reconstruction but also on reconstructing motions.
- [Quality] The method is technically sound. The key idea, as mentioned above, of extending NeRF to the temporal domain, is quite simple and well executed.
- [Clarity] The presentation is overall clear. The core idea as well as the technical details are well presented and easy to follow. Math notations have been used properly.

### Weaknesses
- [Clarity/Contribution] From the current presentation, it is unclear how the proposed work relates to the work of [20]. How does this paper add over [20], and why is [20] not discussed in the related work section?
- [Quality] Although the theoretical framework looks nice, I'm still a little bit skeptical on how the latent space optimization would work practically. It seems that this heavily depends on how smooth the latent space of $z$ is. If the space learned from VAE is not sufficiently smooth, the optimization (e.g. for Eq. 9 and 10) will always be trapped in useless local minimum. Would this be an issue at all? It would be helpful to somehow show the landscape by visualizing the pose transformation as one traverses in the latent space.
- [Clarity] The detail of the generative task for Sec. 4.2.2 and Sec. 4.2.4 is missing. What is the generative task here, i.e. input and output? Is it trying to predict the pose at arbitrary $t$ given some keyframes from an unseen (test) sequence, so the approach is to perform latent space optimization for $z$ and then run forward pass on the evaluated $t$s?

### Justification of Rating
Overall this looks like a solid submission with decent novelty and contribution. I just have some concerns on the comparison with prior work as well as some technical details as listed below.

---

> ### Author Response · Authors · 2022-08-01
> **Response to Reviewer wNce (1/3)**
>
> Dear Reviewer wNce:
>
> Thanks for your detailed and thorough review! We sincerely appreciate the valuable suggestions you provided! In the following, we seek to address your concerns one by one:
>
> **Q:** *“From the current presentation, it is unclear how the proposed work relates to the work of [20]. How does this paper add over [20], and why is [20] not discussed in the related work section?”*
>
> **A:** Sorry that we missed the discussion about HM-VAE [20] in our related work section, we will add it in the future version. To be brief, HM-VAE proposes a convolutional VAE to learn a generic motion prior for multiple downstream applications. Though we have a similar idea in the encoding part, our work takes ideas from implicit neural representations and adopts an MLP as the decoder, while Li et al. [20] just design a convolutional decoder similar to the encoder. As a result, our decoder learns a continuous mapping between the latent space, temporal coordinates, and poses, thus enabling applications such as temporal sub-sampling, which is impossible with HM-VAE.
>
> ---
>
> **Q:** *“It seems that this heavily depends on how smooth the latent space of $z$ is. If the space learned from VAE is not sufficiently smooth, the optimization (e.g. for Eq. 9 and 10) will always be trapped in a useless local minimum. Would this be an issue at all? It would be helpful to somehow show the landscape by visualizing the pose transformation as one traverses in the latent space.”*
>
> **A:** Right. Though the smoothness of our latent space is verified by latent space interpolation, some problems arise when the optimization constraints are too weak (e.g., when trying to fit a motion sequence from only a single pose), where the output tends to have artifacts such as just producing static motion. As we mentioned in Sec. 5.1, other generative architectures such as GAN or Diffusion Model may be required if we would like to address this problem completely. As for visualization, since the latent space we learned is a motion prior, each latent code already encodes an entire motion sequence. Thus, the entire output sequence will change when navigating in the latent space. Here we visualize the latent distribution of our 164 testing sequences by projecting their latent codes into 2D space using UMAP (latent distribution of $z_l$: https://nemf.s3.us-east-2.amazonaws.com/z_l.pdf, latent distribution of $z_g$: https://nemf.s3.us-east-2.amazonaws.com/z_g.pdf). To demonstrate navigation in the latent space, we pick the latent codes we used for latent space interpolation (Fig. 5 in our supplementary) and project them to 2D as well. They are connected and visualized as red points in the previous visualizations to illustrate the trajectory formed in the latent space.
>
> ---
>
> **Q:** *“The detail of the generative task for Sec. 4.2.2 and Sec. 4.2.4 is missing.”*
>
> **A:** We provide details about our motion synthesis task in Sec. 2.3 of our supplementary. To be more specific, we generate 400 samples for each setup by sampling the latent space, so the input is a latent code and the output is a motion sequence. No optimization is used here, the process is just: sample a latent code $z$, run forward pass for each $t$, and obtain the output sequence.
>
> ---
>
> **Q:** *“In general I'm wondering how the latent code handles translation in time. For example, in the current framework, the exact motion clip (e.g. with a time interval [0, T]) after translation (e.g. to interval [-T/2, +T/2]) will have a completely different latent code. This seems a bit awkward and somewhat redundant. I wonder whether there has been any observed pattern on the latent code for translation in time, also, whether one should eventually decouple the translation dimension from the latent code.”*
>
> **A:** First of all, there are two definitions we need to clarify: the first one is relative time, which measures the duration of an event. In our case, that’s the duration of each sequence, which always starts at 0 and ends at 127 (T). The second one is absolute time, which is the timeline that places our motion sequence. Regarding your question, the translation seems to happen in absolute time because relative time in our case will always be [0, T]. If so, time translation will not affect our model because 1) for encoding, only the motion data itself is considered, so the same data will only be mapped to the same latent code no matter where the sequence is placed; 2) for decoding, time steps are considered in relative time to decode poses. After decoding the entire sequence, one can freely put it anywhere in the absolute time axis.
> **Please feel free to comment on whether we understand your question correctly!**

---

> > ### Author Response · Authors · 2022-08-04
> > **Update on the time translation question**
> >
> > Dear reviewer wNce:
> >
> > We have a new perspective to understand your question on the time translation, let’s paraphrase it first: suppose we have a sequence of 300 frames. In our setup, we need to chop a 128-frame clip for encoding and decoding. If clip $A$ contains frames from 1 to 128 and clip $B$ contains frames from 11 to 138, then these two clips share 118 frames, but with a 10-frame offset. Your question is, will the latent codes of clips $A$ and $B$ be completely different?
> >
> > To answer this question, we set up an experiment: we first randomly pick 3 sequences from the AIST++ dataset, each containing about 200 to 300 frames. For each sequence, we use a sliding window with an offset of 10 to obtain clips with 118-frame overlap, which means the first clip is from 1 to 128, the second clip is from 11 to 138, and the third clip is from 21 to 148, etc. We encode these clips using our encoder and then project the corresponding latent codes into 3D space with UMAP. We finally connect each latent code within the same sequence to see if there are some specific patterns. The visualizations are provided at https://nemf.s3.us-east-2.amazonaws.com/z_l_trans.pdf and https://nemf.s3.us-east-2.amazonaws.com/z_g_trans.pdf, where different colors represent latent codes from different sequences (there are 3 in total). In the visualization, we can observe that after the time translation, although their latent codes will be different, they form a circling pattern in the latent space, suggesting that they do remain some connections. **Again, please feel free to comment on whether we understand your question correctly!**

---

> > > ### Comment · Reviewer_wNce · 2022-08-05
> > > **Response to Authors**
> > >
> > > Thanks. This answers my question. It is good to see that translation in time results in similar code in the latent space.

---

> > ### Comment · Reviewer_wNce · 2022-08-05
> > **Response to Authors**
> >
> > **A:** *"First of all, there are two ... anywhere in the absolute time axis."*
> >
> > Thanks for the clarification. I missed the fact that the decoder is trained with $t\in[0, 127]$, so it is expected to only operate in in this interval. This actually makes sense.

---

> ### Author Response · Authors · 2022-08-01
> **Respone to Reviewer wNce (2/3)**
>
> **Q:** *“[Eq. 5] Should there also be a loss term for global root translation? The supplementary Sec. 1.1 mentions a loss function for the global motion predictor (i.e. $\mathcal{L}$  in Eq. 2). Is that also part of $\mathcal{L}_{rec}$ here? If not, how is the global motion predictor loss applied in the framework?”*
>
> **A:** When training our single-motion NeMF model for the sanity test, there is indeed a loss term for the global translation, which is the same as Eq. 2 and 3 in supplementary. In this case, the MLP fits both local and global motion. While in the generative case, we observed some artifacts as illustrated in Fig. 1 of our supplementary, thus, we chose to train a standalone global motion predictor to handle the global motion and dropped the loss term when training the VAE. For consistency, we didn’t mention it in Eq. 5, but we will add this detail to our supplementary in the future version.
>
> ---
>
> **Q:** *"[Tab. 3] How come some of the numbers in "Ours" is exact to those in Tab. 2 (e.g. 5.988 MRE, 2.870 MPE, 6.157 MOE, and 29.692 MTE) while some are different (e.g. 6.508 versus 2.073 FID, 2.118 versus 2.774 Diversity, and 0.566 versus 0.573 FS)?"*
>
> **A:** That's because the pretrained HM-VAE model we use for comparison outputs sequences of 64 frames instead of 128 frames. This is not a problem for motion reconstruction because we can cut the input sequence in half, encode/decode them separately, and concatenate the final output, so the numbers we reported for motion reconstruction are the same. For motion synthesis, however, this is indeed a problem because we cannot concatenate sequences generated from different latent codes. Besides, we cannot evaluate metrics such as FID on sequences of different lengths. Therefore, we chose to cut our and HuMoR’s results in half, so they both have 64 frames. We then evaluate the metrics on these chopped sequences, that’s why the numbers we reported for motion synthesis are different.
>
> ---
>
> **Q:** *"[L171] "In our experiments, $z_l$ and $z_g$ are initialized using the SLERP data to facilitate convergence." -> The details are unclear. Is SLERP first used to perform interpolation in the joint angle space, and then the joint angles are passed through the encoder to obtain the latent code?"*
>
> **A:** Your understanding is correct! We removed this detail due to page limitations, but we can definitely add it to our supplementary.
>
> ---
>
> **Q:** *“[Fig. 2] I wonder whether the absolute velocity values are the best metric to look at for this motion reconstruction evaluation. Would it be better to evaluate on the error (of both position and velocity) rather than the absolute value? The issue may be that it is not possible to get the ground-truth position and velocity at frame rates higher than 30 FPS. Then would it be reasonable to train on 15 FPS and evaluate on 30 FPS then just for this experiment?”*
>
> **A:** The point of using absolute velocity values in Fig. 2 is not to evaluate the reconstruction accuracy, but the smoothness of the sequence. From this point, errors on joint positions may not be a good metric since they cannot provide an intuitive clue about the smoothness.
>
> ---
>
> **Q:** “[L273] "Compared to HuMoR and our model, it is **obvious** that HM-VAE" models an over-smoothed latent space, thus filtering high-frequency details in both the reconstructed and synthesized motions, ..." -> Not so obvious. Following the weakness point for [20] above, it is unclear how [20] works and why it "models an over-smoothed latent space" while HuMoR and "Ours" don't.”
>
> **A:** As we mentioned earlier, [20] proposes a convolutional VAE that learns a generic motion prior. However, in our experiments with their pretrained model, we observed that their reconstruction and optimization results tended to lack dynamic details, which led to the conclusion that their latent space is over-smoothed. This artifact is more pronounced in the motion in-betweening results (Fig. 5), where HM-VAE produced over-smoothed results and failed to reconstruct the given keyframes.
>
> ---
>
> **Q:** *“[L274] "..., making their results worse in both the reconstruction and synthesis task." -> Not precisely. HM-VAE seems to outperform HuMoR sometimes, e.g. on MRE, MPE, MOE?”*
>
> **A:** Thanks for pointing it out! We will change our wording in the future version.

---

> > ### Comment · Reviewer_wNce · 2022-08-05
> > **Response to Authors**
> >
> > **A:**  *"That's because the pretrained HM-VAE model we ... reported for motion synthesis are different."*
> >
> > Thanks for the clarification. One follow up on FID in Tab. 2 and 3: why did the FID metric increase drastically (i.e., from 2.073 to 6.508) when the sequence length is reduced from 128 to 64? Is the FID value expected to be comparable between different sequence lengths?

---

> > > ### Author Response · Authors · 2022-08-06
> > > **Response to the question on FID**
> > >
> > > Dear Reviewer wNce:
> > >
> > > As we wrote on [L130] of our supplementary, we train an autoencoder as the feature extractor to evaluate FID. Since our autoencoder is trained with 128 frames while all data in the comparison are 64 frames, we pad all comparison data with zeros. These data after padding deviates from the manifold spanned by plausible 128-frame motions, that’s why they produce a drastically worse FID.

---

> ### Author Response · Authors · 2022-08-01
> **Response to Reviewer wNce (3/3)**
>
> Regarding your high-level suggestions, we also provide some discussion:
>
> **Q:** *“[High level] So far the paper still utilizes a low dimensional vector at the output for obtaining human pose and shape. How does it compare to making the model "fully implicit", i.e. on not just the temporal but also the spatial domain (x, y, z) as well, like in the original NeRF and other implicit shape models? I guess as the first step, it may just be easier to rely on a forward function to generate the mesh from the pose vector, which is why the authors go in this route first. Yet a "full implicit" model will make the framework look more elegant and potentially better integrated with the efforts on the NeRF side.”*
>
> **A:** That’s definitely an interesting direction! There are already some works that leverage implicit representations to model articulated objects [1, 2, 3, 4], but they still have certain limitations such as not being generalizable. We can explore more in this interesting direction. Besides, although such a fully implicit representation will make it easy for neural rendering like NeRF, these models may not be easy to integrate with current animation workflows (e.g., retargeting, rigging, etc.). How to define a representation that is friendly to both emerging neural rendering and traditional animation workflows is also an interesting direction to try.
>
> ---
>
> **Q:** *“[High level] The paper has shown results on an array of "offline" tasks, where the motion of the full sequence can be pre-solved through a latent code and fixed thereafter. Meanwhile, I would imagine an "online/interactive" variant of the NeMF framework would be useful, especially in many gaming applications (e.g. real-time character control, etc.). Are there ways to adapt the current framework to an interactive setup? Perhaps one can run the latent space optimization and motion generation in a continuous loop by streaming motion generation in short time segments (e.g. 1 sec)? If so, it would be nice to add some mention to such extensions.”*
>
> **A:** That’s also an interesting direction to try! Though as we mentioned in Sec. 2, these interactive applications usually rely on “Time Series Models” that auto-regressively predict future frames, how to integrate our NeMF setup into these interactive applications is definitely interesting. We will mention it as future work in our revision.
>
> **References:**
> 1. Deng, Boyang, et al. "Nasa neural articulated shape approximation." European Conference on Computer Vision. Springer, Cham, 2020.
> 2. Mihajlovic, Marko, et al. "LEAP: Learning articulated occupancy of people." Proceedings of the IEEE/CVF Conference on Computer Vision and Pattern Recognition. 2021.
> 3. Chen, Xu, et al. "SNARF: Differentiable forward skinning for animating non-rigid neural implicit shapes." Proceedings of the IEEE/CVF International Conference on Computer Vision. 2021.
> 4. Noguchi, Atsuhiro, et al. "Neural articulated radiance field." Proceedings of the IEEE/CVF International Conference on Computer Vision. 2021.

---

> > ### Comment · Reviewer_wNce · 2022-08-05
> > **Response to Authors**
> >
> > Thanks for the feedback.

---

### Official Review · Reviewer_umUH · 2022-07-17

**Rating:** 6
**Confidence:** 4
**Soundness:** 3 good
**Presentation:** 2 fair
**Contribution:** 3 good

**Summary:**

The authors propose NeMF (neural motion fields), an implicit neural representation for human motion. Similar to NeRF, they demonstrate that they can well encode a mapping function from time-step `t` to poses. They also learn a generative model by conditioning this on a latent variable `z` to encode different styles/kinds of motion.

They demonstrate that NeMF can well encode a single motion sequence of varying lengths. Their generative samples yield good quality motions, as demonstrated quantitatively and qualitatively compared to previous methods. They also perform a few downstream tasks such as motion in-betweening, spare keyframe in-betweening, and re-navigating tasks by formulating them as a latent space optimization task (find `z` that best approximates the inputs).


**Questions:**

I’m curious to know more about latent-space optimization and convergence. Is the optimization stable, and does it converge when not using custom initialization? Also, for the re-navigating task, is the model able to optimize and generate complex paths (say writing out the word NEURIPS). Some more details on the optimization setup would be helpful.

It’s interesting to note that NeMF has significantly better foot-skate numbers when compared to HuMoR even though HuMoR explicitly models ground contacts. Could the authors comment more on this phenomenon? Is it due to diverging motion?

I think some places in the text are missing adequate explanations and justifications (especially in the methods and experimental section). Some places are:
- Before Eq 2 maybe a line or two saying you are only interested in predicting the joint angles (and not the full X_t as defined previously).
- Eq 7 and 8, it’s unclear why you move back to X and X_t notation for Eq 7. It’s also not clear in Eq 8 what lower-case x is, it’s not defined previously/after the Eq.
- Details/setup of how the AIST++ motion in-betweening was done is missing.
- Typos: Eq 1 should have an angular velocity term (last term)


**Limitations:**

Yes, the authors have included both limitations and broader impact sections in the main paper.


**Strengths And Weaknesses:**

Strengths:
- The problem setup and formulation makes sense. The authors demonstrate good performance in both single motion encoding and the generative model.
- The implicit neural representation has several advantages (as shown by temporal sub-sampling and good motion reconstruction and synthesis results).
- The authors have done several evaluations on different tasks. The supplemental material show convincing qualitative visualizations of the model.

Weakness:

I think the paper suffers a bit from a lack of details and clarity in writing (detailed below).

Otherwise, I think the paper's contributions would be a good addition to the community.

---

> ### Author Response · Authors · 2022-08-02
> **Response to Reviewer umUH**
>
> Dear Reviewer umUH:
>
> Thanks for your detailed and thorough review! We sincerely appreciate your suggestions and pointing out typos. We will revise them in the future version. In the following, we seek to address your concerns one by one:
>
> **Q:** *“I’m curious to know more about latent-space optimization and convergence. Is the optimization stable, and does it converge when not using custom initialization?”*
>
> **A:** Based on our experiments, all the optimizations converge within 600 iterations with custom initialization. When we change to the Gaussian initialization, sometimes the optimization converges at a point that produces results with less dynamic details. This suggests that although the latent space we learned is smooth, it’s not truly Gaussian due to the limitations of VAE. As we mentioned in Sec. 5.1, other generative architectures such as GAN or Diffusion Model may be required if we would like to address this problem completely.
>
> ---
>
> **Q:** *“Also, for the re-navigating task, is the model able to optimize and generate complex paths (say writing out the word NEURIPS).”*
>
> **A:** Redirecting the reference motion to the word “NEURIPS” seems hard because the entire path is too long and complex to complete a sequence of 128 frames. To showcase complex paths, we instead generated 4 paths forming the words “2022”, and re-navigated the walking motion in Fig. 6 to follow these paths. The results are visualized at https://nemf.s3.us-east-2.amazonaws.com/renavi_2022.mp4.
>
> ---
>
> **Q:** *“Some more details on the optimization setup would be helpful.”*
>
> **A:** We provide details about our test-time optimization in Sec. 2.2 of our supplementary.
>
> ---
>
> **Q:** *“It’s interesting to note that NeMF has significantly better foot-skate numbers when compared to HuMoR even though HuMoR explicitly models ground contacts. Could the authors comment more on this phenomenon? Is it due to diverging motion?”*
>
> **A:** Yes, we examined HuMoR’s data we used for evaluation, and after their divergence, there are indeed some motion clips that 1) have less dynamic details, and 2) slide on the ground. These diverging data possibly answer the question of why HuMoR produces a worse foot-skate number.
>
> ---
>
> **Q:** *“Eq 7 and 8, it’s unclear why you move back to $\mathbf{X}$ and $\mathbf{X}_t$ notation for Eq 7. It’s also not clear in Eq 8 what lower-case $\mathbf{x}$ is, it’s not defined previously/after the Eq.”*
>
> **A:** In Eq. 7 and 8, we omit $t$ because $\mathbf{X}$ represents the input to the encoder, which covers the entire sequence instead of only a single frame (as we wrote on [L147]). In Eq. 8, the lower-case $\mathbf{x}$ is actually a typo, which should be $\mathbf{x}^r$, the joint rotations outputting from the decoder (defined on [L150]). Again we omit $t$ here because it represents joint rotations covering the entire sequence.
>
> ---
>
> **Q:** *“Details/setup of how the AIST++ motion in-betweening was done is missing.”*
>
> **A:** Sorry that we forgot to mention it in our paper. When doing motion in-betweening for AIST++, we first randomly pick two videos with their fitted SMPL parameters. We then chop 22 frames from the first sequence and 76 frames from the second sequence, and finally set the transition length to 30 frames (thus 128 frames in total). After that, we run our latent optimization method to find the latent code and decode it to obtain the motion that fills the gap. We will put these details into our paper or supplementary in the future version.

---

### Meta-Review · Area_Chair_j6rk · 2022-08-24

**Recommendation:** Accept
**Confidence:** Certain

**Metareview:**

All reviewers are in favor of accepting the submission post rebuttal.  The AC agrees.   Please further revise the paper based on the reviews, in particular the very constructive comments from reviewer wNce.

**Award:**

No

---

### Decision · Program_Chairs · 2022-09-14

Accept